# Barriers and Facilitators to Implementing Reduced-Sodium Salts as a Population-Level Intervention: A Qualitative Study

**DOI:** 10.3390/nu13093225

**Published:** 2021-09-17

**Authors:** Xuejun Yin, Maoyi Tian, Lingli Sun, Jacqui Webster, Kathy Trieu, Mark D. Huffman, J. Jaime Miranda, Matti Marklund, Jason H. Y. Wu, Laura K. Cobb, Hongling Chu, Sallie-Anne Pearson, Bruce Neal, Hueiming Liu

**Affiliations:** 1The George Institute for Global Health, University of New South Wales, Sydney, NSW 2050, Australia; xyin@georgeinstitute.org.au (X.Y.); jwebster@georgeinstitute.org.au (J.W.); ktrieu@georgeinstitute.org.au (K.T.); mark.huffman@nm.org (M.D.H.); jaime.miranda@upch.pe (J.J.M.); mmarklund@georgeinstitute.org.au (M.M.); jwu1@georgeinstitute.org.au (J.H.Y.W.); bneal@georgeinstitute.org.au (B.N.); hliu@georgeinstitute.org.au (H.L.); 2The George Institute for Global Health, Peking University Health Science Center, Beijing 100600, China; lsun1@georgeinstitute.org.cn; 3School of Public Health, Harbin Medical University, Harbin 150081, China; 4Feinberg School of Medicine, Northwestern University, Chicago, IL 60611, USA; 5CRONICAS Centre of Excellence in Chronic Diseases, Universidad Peruana Cayetano Heredia, Lima 15074, Peru; 6Department of Medicine, School of Medicine, Universidad Peruana Cayetano Heredia, Lima 15074, Peru; 7Department of Epidemiology, Johns Hopkins Bloomberg School of Public Health, Baltimore, MD 21205, USA; 8Department of Public Health and Caring Sciences, Uppsala University, 75236 Uppsala, Sweden; 9Resolve to Save Lives, an Initiative of Vital Strategies, New York, NY 10005, USA; lcobb@resolvetosavelives.org; 10Research Center of Clinical Epidemiology, Peking University Third Hospital, Beijing 100083, China; chuhl_pucri@bjmu.edu.cn; 11Centre for Big Data Research in Health, University of New South Wales, Sydney, NSW 2052, Australia; sallie.pearson@unsw.edu.au; 12School of Public Health, Imperial College London, London SW7 2BX, UK; 13Sydney Institute for Women, Children and Their Families, Sydney Local Health District, Sydney, NSW 2050, Australia

**Keywords:** reduced-sodium salt, salt substitute, low-sodium salt, sodium reduction, potassium, key informant interview, qualitative study

## Abstract

Widespread use of reduced-sodium salts can potentially lower excessive population-level dietary sodium intake. This study aimed to identify key barriers and facilitators to implementing reduced-sodium salt as a population level intervention. Semi-structured interviews were conducted with key informants from academia, the salt manufacturing industry, and government. We used the reach, effectiveness, adoption, implementation, and maintenance (RE-AIM) framework to inform our interview guides and data analysis. Eighteen key informants from nine countries across five World Health Organization regions participated in the study from January 2020 to July 2020. Participants were concerned about the lack of robust evidence on safety for specific populations such as those with renal impairment. Taste and price compared to regular salt and an understanding of the potential health benefits of reduced-sodium salt were identified as critical factors influencing the adoption of reduced-sodium salts. Higher production costs, low profit return, and reduced market demand for reduced-sodium salts were key barriers for industry in implementation. Participants provided recommendations as potential strategies to enhance the uptake. There are presently substantial barriers to the widespread use of reduced-sodium salt but there are also clear opportunities to take actions that would increase uptake.

## 1. Introduction

High blood pressure is a major risk factor for cardiovascular disease and premature death worldwide [1]. Higher sodium intake and inadequate potassium intake are associated with elevated blood pressure and risk of cardiovascular diseases (CVDs), whereas higher potassium intake is associated with reduced blood pressure [2,3].

The World Health Organization (WHO) recommends a dietary sodium intake for adults of less than 2 g per day and a potassium intake of at least 3.5 g per day [4]. Whilst there is debate about the lower threshold of sodium intake [5], most of the world’s population consumes more sodium than the current recommendation [6]. Different population-level strategies are initiated by governments and authorities to reduce dietary sodium and increase dietary potassium consumption [7]. The United Kingdom (UK) has successfully implemented a population salt reduction programme, leading to a 15% reduction in the average sodium intake of the population over 7 years. The reduction of the UK population’s sodium intake was mainly achieved by working with the food industry on gradual reformulation of packaged foods on a voluntary basis [8]. However, this programme may not have the same impact in East, Central, and South Asia and many low- and middle-income countries where most sodium comes from salt added during cooking [9,10]. An alternative intervention is therefore required in these settings to reduce the use of excessive sodium consumption. Reduced-sodium salts with added potassium have the promise, as a population intervention, to meet the WHO sodium and potassium targets, and subsequently reduce the disease burden attributable to high blood pressure [11].

Regular salt is about 98%–100% sodium chloride (NaCl). Reduced-sodium salts, also known as salt substitutes, are table or cooking salts that have reduced (12–100% less) sodium chloride content through the substitution with potassium chloride or other minerals. Replacing regular salt with reduced-sodium salt can reduce sodium intake and increase potassium intake. The effect of reduced-sodium salts on mineral intake and blood pressure, the risk of stroke, and major cardiovascular events has been demonstrated in controlled research settings [12,13,14]. There may be population-level benefits from widespread reduced-sodium salt consumption achieved through reducing hypertension and ultimately cardiovascular disease. A comparative risk assessment model estimated recently that nationwide replacement of the salt supply with potassium-enriched, reduced-sodium salt in China would prevent one in every nine CVD deaths [15]. A recent review has recommended population uptake of reduced-sodium salt as one of the priority strategies to reduce population sodium consumption [16]. A meta-analysis found that, among four salt reduction interventions conducted in China with high quality randomised control trial design, only reduced-sodium salt was demonstrated as an effective intervention to lower blood pressure [17]. A recent research study showed that a range of modelled reduced-sodium salt strategies had the potential to be cost-effective in lowering blood pressure at a population level in Vietnam [18]. Despite reduced-sodium salts being effective and cost-effective, as demonstrated in trials and modelling studies, reduced-sodium salts are not widely available on the market and are more expensive than regular salts [19].

The uptake of reduced-sodium salt as a population health intervention requires a deeper understanding of factors influencing its adoption, implementation, and maintenance. Existing literature about reduced-sodium salts has mainly focused on their effectiveness, with limited evidence to inform the factors influencing the widespread use of reduced-sodium salt in the general population. This qualitative study aims to address this gap and understand the barriers and opportunities with a view to better define the potential for scale-up of reduced-sodium salt interventions as a population health intervention.

## 2. Materials and Methods

This study is part of a broader environmental scan, consisting of a systematic online search, systematic review, and key informant interviews. The protocol of the environmental scan has been published [20]. The study received ethics approval from the University of New South Wales Human Research Ethics Advisory Panel (HC190921).

### 2.1. Key Informant Identification and Recruitment

We included key informants who have been involved in the manufacturing, research, implementation, or promotion of reduced-sodium salts. We used purposive and snowball sampling to recruit interview participants. Academic representatives were identified through the systematic review component of the environmental scan. Corresponding authors of the included studies were contacted by email to confirm their willingness to participate. Academic representatives who participated in the interview were asked to pass on the recruitment materials and share the investigator’s contact information to potential key informants from relevant government agencies or the salt manufacturing industry. Of note, helping to contact other key informants was not a requisite for participating in the research. These key informants from relevant government agencies or the salt manufacturing industry could then contact the investigator if they were interested in the study. All key informants who contacted us subsequently agreed to participate. The final sample size of the key informants was determined by data saturation until no new themes emerged [21].

### 2.2. Data Collection

In-depth interviews were conducted between January 2020 and July 2020 by a qualitative researcher (XY). Interviews were conducted mainly in English. For key informants from China who preferred to speak Chinese, interviews were conducted in Chinese, as XY is fluent in Chinese as well. Interviews were conducted either over telephone or video conference, or at the interviewees’ workplace. Informed consent was obtained verbally at the commencement of each interview. Consent included permission to be audio-recorded.

Semi-structured interview guides were designed based on the domains of the reach, effectiveness, adoption, implementation, and maintenance (RE-AIM) framework [22]. The RE-AIM framework has been widely used for planning and evaluation of health programs and policies. While the full use of the RE-AIM framework includes both qualitative and quantitative components in the current study, we applied RE-AIM qualitatively to focus on for whom, how, and why reduced-sodium salts may be used [23]. Example questions for assessing each dimension are shown in Table 1. The interview guides (Appendix A) were adapted to the different roles of the key informants.

### 2.3. Data Analysis

The audio-recorded interviews were transcribed verbatim. Data analysis was conducted in either English or Chinese [24]. Quotations in Chinese were translated into English and presented in English. This process involved forward-translation (from the source language into English) and back-translation (from English to their source language). Both versions were compared to check accuracy and equivalence. Discrepancies that occurred during the process were discussed between the two bilingual researchers (XY, LS). Both inductive and deductive methods were adopted to analyse the interview transcripts. The headings of RE-AIM were used as an initial framework (deductive approach), and thematic analysis was conducted to generate codes linking to the relevant dimension in the RE-AIM framework (inductive approach). The first round of the analysis involved the first author (XY) reviewing the transcripts verbatim and inductively assigning codes to emergent concepts. The second round of the analysis used the same coding framework to apply codes to the transcripts by another coder (LS) independently. The codebook was generated using constant comparison across the different key informants’ perspectives by which each interpretation and finding was compared with existing findings as it emerged from the data analysis. Coding discrepancies were discussed with a senior researcher (HL) and resolved by consensus to optimise inter-coder reliability. Revisions to the coding scheme were applied to all previously coded transcripts. The NVivo analytical software system (version 12.0) was used for data management and data coding. The study was reported according to the Consolidated Criteria for Reporting Qualitative Research (COREQ) [25].

## 3. Results

Eighteen key informants from nine countries participated in the interview. Table 2 provides an overview of participant characteristics. Among all participants, nine were academic representatives, seven were government representatives, and two were salt manufacturing industry representatives. Twelve participants were from high-income countries, while six were from middle-income countries, with representation across five out of six WHO regions.

Table 3 summarises identified facilitators and barriers to implementing reduced-sodium salt as a population-level intervention based on the RE-AIM domains. Findings were aggregated for each RE-AIM dimension around barriers and facilitators.

### 3.1. Reach

All participants indicated that the goal of promoting reduced-sodium salt was to reach a wider population rather than limit it to hypertensive patients. Many participants described that the use of reduced-sodium salt was relevant for populations whose sodium sources mostly came from home cooking. One participant noted that the use of affordable reduced-sodium salts reaching the entire population regardless of socio-economic status could improve health equity in resource-poor settings, where the main source of sodium is salt adding during home cooking and CVD burdens are high.

“Reduced-sodium salts from my perspective is an intervention which is particularly suited to many of the poorest and least developed communities in the world because their sodium source is mainly from salt they added at the time of cooking or food preparation. This is fantastic for health equity.”(Australia, academic representative)

However, the main perceived barrier to reaching a wider population was the limited availability of reduced-sodium salt in most countries. Even in countries where reduced-sodium salts were available, participants highlighted that sales of reduced-sodium salts (market share) were small. An interviewee from the Chinese salt manufacturing industry reflected that the market share of reduced-sodium salts was less than 10% of the total salt market, with products mainly available in urban areas. An interviewee from Singapore reflected that the market share of reduced-sodium salts was only about 2%. The lack of availability of reduced-sodium salt in rural areas was also reflected by interviewees from Peru and India.

### 3.2. Effectiveness

High-quality evidence of the efficacy and effectiveness of reduced-sodium salts on reducing sodium intake, increasing potassium intake, lowering albuminuria, and lowering blood pressure in trial settings were identified as facilitators. However, uncertainty about its real-world effectiveness of reducing sodium intake and lowering blood pressure at a population-level and concern about safety in patients with renal disease were identified as barriers to scale up. Participants also highlighted that people might add more reduced-sodium salt to maintain the same salty taste of food. Behavioural interventions, such as health education explaining the need to reduce excessive sodium intake coupled with the use of reduced-sodium salt, were suggested to enhance the effectiveness of reduced-sodium salt at the population-level.

Participants identified that it was important to educate the general population about the appropriate use of reduced-sodium salt in two key areas. First, most reduced-sodium salts still contain a significant proportion of sodium chloride (0–88%). Education on consuming less reduced-sodium salt is suggested. Second, despite reduced-sodium salts being effective in lowering the blood pressure of hypertensive patients, adherence to anti-hypertensive medications is still required.

“Using reduced-sodium salts doesn’t mean people will reduce sodium intake, and this is something that should be further investigated.”(Italy, salt manufacture industry representatives)

Participants reinforced that more research is required on the effectiveness and safety of reduced-sodium salts for patients with renal disease. Two researchers confirmed that using reduced-sodium salts can decrease albuminuria level and prevent further adverse impacts of renal disease. Yet, there was an evidence gap in terms of the long-term safety and effectiveness of reduced-sodium salts in patients with renal disease.

“I am convinced it is beneficial for albumin reduction. Albumin is a marker of the problems of kidney. After decreasing the albumin level, people will be better off, if there is a longer study that can see the reduction in the progression of kidney diseases that would be a major achievement.”(Australia, academic representatives)

Although participants from salt companies described being confident in the safety of reduced-sodium salts, and researchers advocated the substantial net benefits of reduced-sodium salts, government representatives expressed ongoing concerns about its safety as the key barrier to scaling up reduced-sodium salt as a population level intervention.

“There is much evidence for the benefits of reduced-sodium salt. We actually have considered it as one of the salt reduction strategies. However, safety is a challenge when taking it as a public health intervention. Reduced-sodium salt contains potassium chloride; a tiny number of patients should not use reduced-sodium salt. Once a case of hyperkalemia caused by reduced-sodium salt is reported it will cause trouble. From the perspective of policymakers, we take risks and responsibilities. I can advocate (for) reduced-salt, but I would not recommend reduced-sodium salt to the public.” (China, Government)

### 3.3. Adoption

Key factors influencing the adoption of reduced-sodium salts include higher prices, different taste, and knowledge, attitude, and behaviour towards reducing excessive sodium intake. Participants from low- and middle-income countries highlighted the importance of affordability of reduced-sodium salts. One participant from China commented that while the price difference was an important barrier to those living in rural areas, it was not the primary barrier for people in purchasing reduced-sodium salt in cities. He highlighted that, although the price of reduced-sodium salts was about one and a half times that of regular salt, it was still affordable to urban residents but that they generally had limited awareness of its availability and health benefits. In addition, it was also highlighted that consumers may not like the taste. Potassium chloride tastes both salty and bitter compared to the pure salty taste of sodium chloride. Researchers suggested that the sodium level in the reduced-sodium salts can be reduced gradually by the food industry over time to increase acceptability of the taste. Participants suggested more research related to the acceptability of the taste of reduced-sodium salts using different formulas of sodium and potassium are needed.

“We were trying to be careful about the acceptance levels that will have an impact on their options. You need to figure out rather than assuming what sodium level of salt will work. I rather do it gradually. I think the ideal behaviour is to cook with the same or less amount of reduced-sodium salts when cooking. If the sodium changed dramatically, people would end up thinking this is not salt. I won’t use it when cooking.”(Peru, academic representatives)

### 3.4. Implementation

Higher manufacturing costs and reduced profit margins influenced the production of reduced-sodium salt. According to one salt manufacturing industry representative, food companies have made limited investment in reduced-sodium salts because of limited consumer demand and higher manufacturing costs. Moreover, the salt manufacturing industry is required to invest in research and development in salt processing technology to improve the taste of reduced-sodium salts. Participants from salt companies described that short and long-term profit returns need to be considered prior to the manufacturing of reduced-sodium salts.

“We went to a salt company. They told us that they were not interested in reduced-sodium salt. When understanding why, we found out that their business was concentrated on the production of salt for industrial purposes, that salt for human consumption was a minor part of their business and, reduced-sodium salts were less of a priority. How to switch this company to produce more quantities at a reduced price? The answer is to produce a lot. It will be cheaper when mass production becomes possible.”(Peru, academic representative)

Intensive health education programmes to increase consumer demand, reducing the retail price by mass production, mass media campaigns, and incentivizing food industries were identified as potential implementation strategies to increase uptake. Ongoing health education was suggested by all participants, especially education targeted to stakeholders in settings such as schools, restaurants, community centres, and hospitals as well as the general population. Reducing the retail price through government subsidies was regarded as a strong intervention in underdeveloped areas where people were more price sensitive. Mass media campaigns by government were suggested as a critical channel to effectively disseminate information to enhance the use of reduced-sodium salts. In addition, the food industry also needed to be informed and incentivised so that it could recognise the public health importance of reducing sodium levels in their products and how reduced-sodium salts could help them reformulate their products.

### 3.5. Maintenance

The market share of the reduced-sodium salt was highlighted as an important factor for maintenance and sustainment as a population-level intervention in competitive market forces partly due to technological investment costs. An example was reported from China, there were efforts to implement reduced-sodium salt intervention at the population level in several provinces, at the time when salts were only produced by the national salt monopoly, China Salt, a state-owned enterprise that could decide on the price and the distribution of products. This monopoly approach had great potential to increase the reduced-sodium salt market share; however, after the salt market reform in 2017, the market share of the reduced-sodium salt significantly dropped. The key principle of the salt market reform was to open the market to private salt companies, bringing numerous, smaller salt companies into the market, that were less interested in or were not capable of producing reduced-sodium salt because of its higher cost and more complex technology requirements.

“Before the reform in 2017, all salt was produced by the provincial salt company. If they wanted to promote reduced-sodium salt, they would produce more reduced-sodium salt and put it in a prominent place on the shelf in supermarkets. After the reforms in 2017, private salt companies entered the market, and state-owned salt companies could no longer control the market, and the market of reduced-sodium salt significantly dropped.”(China, government)

## 4. Discussion

This research identified significant barriers to the widespread use of reduced-sodium salt, but also clear opportunities to take actions that would increase uptake. A key finding was that, while stakeholders believed there was strong evidence of the effectiveness for lowering blood pressure, there were concerns about safety. Specifically, the possibility that if reduced-sodium salt with added potassium was used in the general population, the subset of persons with severely impaired renal function might be at risk of hyperkalaemia and sudden cardiac death [26].

Government representatives, in particular, were cautious about recommending reduced-sodium salts as a population-wide intervention in light of this potential safety issue. A safety trial on reduced-sodium salts demonstrated that a considerable part of the sodium in regular salt could be replaced by potassium and magnesium salts without causing potassium or magnesium toxicity [27]. Modelling studies in both Chinese and UK populations showed the potential benefits of using reduced-sodium salt substantially outweigh the potential risks [15,28]. Such data could potentially help to provide evidence-informed metrics about safety and ensure that salt labelling correctly addresses the likely balance of risks and benefits associated with the use of reduced-sodium salts in different population groups.

In addition, health education was identified as critical to improving the general population’s knowledge, attitude, and behaviour to reduced-sodium salt. Recent baseline surveys performed for the Salt Substitute and Stroke Study (SSaSS) in China reported the awareness of reduced-sodium salts was only 5.9%. Similarly, no participants had heard about reduced-sodium salts at baseline in the Salt Substitute in India Study (SSiIS) [29,30]. While not representative of the population across the world, findings from these studies suggest an important opportunity for raising awareness of reduced-sodium salt. Participants in this study reflected awareness and health education campaigns are needed in conjunction with reduced-sodium salt to increase public awareness and prompt appropriate use of reduced-sodium salts. Several trials of reduced-sodium salt intervention have showed an increase in mean potassium intake and a decrease in blood pressure with no change in sodium intake levels [31,32]. These results indicated that the reduced-sodium salt intervention was accepted and adopted, but participants were likely using a greater quantity of reduced-sodium salt than regular salt when cooking and eating. Despite of this, replacing regular salt with reduced-sodium salt when cooking can lead to a decreased blood pressure because of the changed sodium/potassium ratio [33].

The higher price of reduced-sodium salt as compared to regular salts was identified as one of major barriers, particular in rural areas. A cluster randomised control trial of salt reduction conducted in rural China showed that adoption of reduced-sodium salts was much higher in villages where the price of reduced-sodium salts was subsidised to be equal to the cost of regular salt compared to villages without a price subsidy [34]. Participants in the present study highlighted that delivery of health education to socioeconomically disadvantaged populations would likely be insufficient and would be significantly enhanced by fiscal incentives, such as government subsidies that remove the price differential to regular salt [35].

Additionally, creating supportive environments for implementing reduced-sodium salt interventions will be important [36]. A collaborative approach involving partnership across government, food industry, and individuals is likely to optimise the opportunity for reduced-sodium salt interventions. For example, governments will need to set sodium reduction targets for food providers, but industries need to provide the reduced-sodium salt for food preparation, and consumers will need to find the products acceptable. Existing studies have shown that using reduced-sodium salts in bread, pizza, and noodles is feasible and widely accepted among consumers, without compromising taste [37,38,39,40]. Reduced-sodium salts could potentially be incentivised for use in restaurants, schools, hospitals, and public canteens and an implementation trial has showed the effectiveness of a reduced-sodium salt intervention in nursing home canteens [41].

This study’s strengths include the depth of data collection, the use of an established implementation science framework and the collection of perspectives from multiple key informants worldwide. The RE-AIM framework helped structure key informant interviews to examine the feasibility and facilitators of implementing reduced-sodium salt in different contexts. On the other hand, the current study also has some limitations. First, data were collected from key informants in only nine countries which may limit the generalisability of the study findings, although representation did include five out of six WHO regions. Second, the diversity of key informants was limited, with a preponderance of academics. While the inclusion of more academics was driven by the mode of sampling, all potential barriers and opportunities may not have been captured from less well represented groups. In particular, achieving comprehensive input from the salt manufacturing industry was difficult because contacts and interviews were difficult to obtain. Some second-hand information about industry perspectives was gleaned from other informants who had interactions with the salt manufacturing sector. We elected not to include consumers in the scope of this project because the views of consumers have been well documented in prior qualitative and quantitative research projects [42].

## 5. Conclusions

Guided by the RE-AIM framework, this study provided new information about facilitators and barriers to implementing reduced-sodium salt interventions from a population-wide public health angle. Reduced-sodium salts appear to have significant potential as a population-level salt reduction intervention, particularly for people whose dietary sodium mainly comes from cooking and food preparation in the home. There is, however, controversy regarding scaling-up reduced-sodium salt as a population intervention due to the uncertainty about safety among patients with serious renal disease. This limits the acceptability of this intervention option to policymakers. The participants highlighted that health education may increase consumer demand for these products, and together with policy incentives that target industry, it may be possible to increase profitability from production and reduce costs to the consumer. Long-term efforts addressing community awareness and industry barriers, in conjunction with better data about the safety of reduced-sodium salts, are needed to successfully develop and implement reduced-sodium salts as a meaningful population health intervention. In conclusion, despite several barriers to the widespread use of reduced-sodium salt, the uptake of reduced-sodium salt can be increased by improving its availability, increasing the awareness of its benefit, addressing concerns about safety, and lowering its price on the market. The important next step is to develop, in conjunction with local public health decision makers, feasible implementation strategies that could address these barriers and integrate reduced-sodium salts as part of the overall public health intervention, based on further investigations of local context.

## Figures and Tables

**Table 1 nutrients-13-03225-t001:** The reach, effectiveness, adoption, implementation, and maintenance (RE-AIM) framework (RE-AIM) dimensions and qualitative data example questions.

Dimension	Example Questions
Reach	To what extent do you think reduced-sodium salts are likely to reach the target population for sodium reduction? How will the access to reduced-sodium salt be supported?
Effectiveness	What do you perceive as the key benefits of reduced-sodium salt? Do you think reduced-sodium salt can be an effective approach to reduce population sodium intake? Why or why not? Prompt: Are there other strategies that you think would be more appropriate? If so, what?
Adoption	How would you improve consumers’ adoption of reduced-sodium salts? What do you think are the main barriers to consumers’ adoption of reduced-sodium salts? Prompt: What suggestions do you have on how to overcome those barriers?
Implementation	How would you implement a reduced-sodium salt intervention as a population health intervention? What are the barriers and facilitators? Prompt: Can you describe any related policies, health education campaigns, or industry engagement?
Maintenance	What is needed to maintain a reduced-sodium salt intervention in the long-term? Can you describe any experiences you have had trying to scale up reduced-sodium salts? Prompt: How are these interventions maintained? What are the reasons for the continuing or/not continuing?

**Table 2 nutrients-13-03225-t002:** Participant characteristics.

Characteristics	Academic Representative	Government Representative	Salt Manufacturing Industry Representative
Sex			
Female	5	5	
Male	4	2	2
Country income levels			
High-income	7	4	1
Low-or middle income	2	3	1
WHO regions			
South-East Asia region		1	
European region	1	1	1
Region of the Americas	2	2	
African region	1	1	
Western Pacific region	5	2	1

**Table 3 nutrients-13-03225-t003:** Themes under RE-AIM domains by facilitators and barriers.

Domain	Barriers	Facilitators
Reach	Limited availability of reduced-sodium salts	
Effectiveness	Uncertainty of effectiveness about reduced-sodium salt intervention in reducing sodium intake and lowering blood pressure at a population-level Evidence gap in health impact among patients with renal disease Possibility that a greater amount of reduced-sodium salt would be used compared to normal salt, potentially reducing its effectiveness	Strong efficacy/effectiveness evidence from trials
Adoption	Higher price compared to regular salt Different taste to regular salt Lack of knowledge of its health benefit Unaware of its availability	
Implementation	Higher cost of production Reduced profit return Low demand for reduced-sodium salts	Reduce retail prices Health education tailored to different people/organisations Mass media campaigns
Maintenance	Decreasing trends of reduced-sodium salt market share relative to other salts (regular salt, sea salt, bamboo salt, etc.)	

## Data Availability

The data presented in this study are available on request from the corresponding author. The data are not publicly available due to privacy restrictions.

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
