# Peer review of "Barriers and Facilitators to Implementing Reduced-Sodium Salts as a Population-Level Intervention: A Qualitative Study"

_nutrients, 2021, doi:10.3390/nu13093225_

Round 1

Reviewer 1 Report

Interesting paper related to the continuing "salt hesitancy" in the public health community. The arguments of the pros and cons are presented and discussed. However, there are no firm conclusions or, more reasonably, sound advice on the next steps. It would be useful, for example, to identify which study would be necessary to convince the various agents. 

Author Response

We appreciate the coment from reviewers. We have carefully responded to each comment followed by itemized responses in bold italics.

Reviewer 1:

Interesting paper related to the continuing "salt hesitancy" in the public health community. The arguments of the pros and cons are presented and discussed. However, there are no firm conclusions or, more reasonably, sound advice on the next steps. It would be useful, for example, to identify which study would be necessary to convince the various agents.

Reply: We added advice on the next steps in the conclusion ”In conclusion, despite there were several barriers to the widespread use of reduced-sodium salt, the uptake of reduced-sodium salt can be increased by improving the availability, increasing the awareness of its benefit, addressing concerns about safety, and lowering its price in the market. The important next step is to develop in conjunction with local public health decision makers, feasible implementation strategies that could address these barriers and integrate reduced-sodium salts as part of the overall public health intervention, based on further  investigations of local context.” (Page 11, line 400-407)

 Reviewer 2 Report

The present original work adds an important contribution to the decision of potential strategies to reduce the consumption of salt, precisely with its replacement by the consumption of salt with reduced sodium content.

The work is well designed and presents the results appropriately, however the authors should clarify the following points:

1 - The study includes 9 countries, however only 7 government representatives. It is not clear if these 7 gov representatives are from 7 different countries and if they are, why the only 7 agree to participate? The other 2 from other countries have not agree to participate?

2 - Is not clear how many government representatives and how many salt manufacturing industry representatives are invited to participate, and agree or reject to participate. The authors must clarify this point.

3 - Line 172 to 179 - one participant argues that the use of reduced sodium salt could improve health equity issues in resource poor settings, and the authors concludes that the main perceived barrier was the limited availability, however, is important considers also the higher cost of these salts in this dimension because the higher economic cost implies that only those with greater economic power will have access to these products, increasing social inequities. Other measures like consume less salt in this line of thought, are more suitable to improve health equity issues in resource poor settings. The authors could clarify this point.

Author Response

We appreciate the reviewer's comments. 

We have carefully responded to each comment followed by itemized responses in bold italics.

Reviewer 2 The present original work adds an important contribution to the decision of potential strategies to reduce the consumption of salt, precisely with its replacement by the consumption of salt with reduced sodium content.

The work is well designed and presents the results appropriately, however the authors should clarify the following points:

1 - The study includes 9 countries, however only 7 government representatives. It is not clear if these 7 gov representatives are from 7 different countries and if they are, why the only 7 agree to participate? The other 2 from other countries have not agree to participate.

We updated the Table 2 to make the distribution of different types of key informants clearer. (Page 5, line 171, Table 2) Seven government representatives were from five countries across five WHO regions (WHO Africa Region, WHO Western Pacific Region, WHO South-East Asia Region, WHO America Region and WHO Europe Region),  including both high-income countries and low-or middle-income countries. This sample size has allowed us to reach data saturation.

Conforming to sampling and recruitment process, the number of government representatives invited and reasons for rejection were not recorded. According to the ethics requirement, the recruitment process was such that the academia representatives were asked to pass on study recruitment materials and share the investigator’s contact information to potential key informants from relevant government agencies or the salt manufacturing industry. Of note, helping to contact other key informants was not a requisite for participating in the research.  These key informants from relevant government agencies or the salt manufacturing industry could then contact the investigator if they were interested in the study. All key informants who contacted us subsequently agreed to participate. We have added this in the “2.1 Key informant identification and recruitment” part. (Page 3, line 119-124)

 2 - Is not clear how many government representatives and how many salt manufacturing industry representatives are invited to participate and agree or reject to participate. The authors must clarify this point.

As above, we have added the recruitment process in the “2.1 Key informant identification and recruitment” section to make our recruitment processes clearer, and that we did not know how many government and salt manufacturing industry representatives the academic representatives forwarded our study information to.  (Page 3, line 117-122)

 3 - Line 172 to 179 - one participant argues that the use of reduced sodium salt could improve health equity issues in resource poor settings, and the authors concludes that the main perceived barrier was the limited availability, however, is important considers also the higher cost of these salts in this dimension because the higher economic cost implies that only those with greater economic power will have access to these products, increasing social inequities. Other measures like consume less salt in this line of thought, are more suitable to improve health equity issues in resource poor settings. The authors could clarify this point.

We agreed that price of reduced-sodium salts may affect equitable uptake. We have clarified in this dimension that the participant had described that the use of affordable reduced-sodium salts could improve health equity in resource-poor settings, where the main source of sodium  is salt adding during home cooking and CVD burdens. (Page 6, line 187-191) In our discussion (lines 340-343 , and 350-358), we highlighted how other measures such as health education campaigns are needed in conjunction with the use of reduced sodium salt, and discuss how government subsidies may be required to remove the price differential to regular  salt in resource poor settings.